# Disparities in the Cardiometabolic Impact of Adiposity among African American and Hispanic Adolescents

**DOI:** 10.3390/nu16183143

**Published:** 2024-09-18

**Authors:** Pedro A. Velásquez-Mieyer, Ramfis Nieto-Martinez, Andres E. Velasquez, Xichen Mou, Stephanie Young-Moss, Jeffrey I. Mechanick, Cori Cohen Grant, Claudia P. Neira

**Affiliations:** 1Lifedoc Health, Memphis, TN 38115, USA; pvelasquez@lifedochealth.org (P.A.V.-M.); nietoramfis@gmail.com (R.N.-M.); cneira@lifedochealth.org (C.P.N.); 2Lifedoc Research, 6625 Lenox Park Drive, Suite 205, Memphis, TN 38115, USA; 3Departments of Global Health and Population and Epidemiology, Harvard TH Chan School of Public Health, Harvard University, Boston, MA 02115, USA; 4School of Public Health, University of Memphis, Memphis, TN 38152, USA; xmou@memphis.edu; 5Independent Researcher, Indianapolis, IN 47906, USA; hello@drstephanieyomo.com; 6The Marie-Josée and Henry R. Kravis Center for Cardiovascular Health at Mount Sinai Heart, Division of Endocrinology, Diabetes and Bone Disease, Icahn School of Medicine at Mount Sinai, New York, NY 10029, USA; jeffreymechanick@gmail.com; 7Department of Preventive Medicine and Tennessee Population Health Consortium, The University of Tennessee Health Science Center, Memphis, TN 38103, USA; cgrant1@uthsc.edu

**Keywords:** pediatric, obesity, cardiometabolic, risk factors, adiposity, racial disparities, chronic disease

## Abstract

As adiposity increases in youth, so does the prevalence of cardiometabolic risk factors (CMRFs). The etiology of adiposity-based chronic disease and CMRFs includes ethnoracial disparities that are rarely considered in current treatment approaches. Precision interventions require further characterization of these disparities among high-risk youth. The objective of this study was to characterize differences in CMRF among African American (AA) and Hispanic (H) adolescents with varying levels of adiposity. A cross-sectional analysis of 2284 adolescents aged 12–17 was conducted using 3-year clinical data from Lifedoc Health. CMRF prevalence were compared using χ^2^, with logistic regression models (LRM) applied to explore the relationships between exposures (age, sex, ethnoracial group, adiposity) and CMRF outcomes. Prevalence of CMRF rose with increasing adiposity, which was the strongest determinant of risk overall. However, individual risk profiles differed between the two groups, with H having higher prevalence of metabolic syndrome (MetS), higher triglycerides and liver enzymes, and low high-density lipoprotein cholesterol (HDL-c). Meanwhile, AA had higher prevalence of elevated blood pressure (BP) in the overweight category, prediabetes in overweight to severe obesity, and type 2 diabetes in obesity. LRM showed 3.0-fold greater chance of impaired glucose metabolism in AA than H, who were 1.7, 5.9, and 8.3 times more likely to have low HDL-c, high liver enzymes, and high triglycerides, respectively. Overweight/obesity prevalence was very high among AA and H adolescents. Excess adiposity was associated with an increased prevalence of CMRF, with individual risk factors differing between groups as adiposity increased. Research within routine clinical settings is required to better characterize these discrepancies and ameliorate their adverse impact on health in the transition to adulthood.

## 1. Introduction

The rate of obesity among children and adolescents aged 10–17 years in the US was reported to be 16.2% in the 2019–2020 National Survey of Children’s Health [1]. The prevalence of overweight/obesity varies between ethnoracial groups. Data from the National Health and Nutrition Survey (NHANES) (2015–2016) show that rates in African American and Hispanic children/adolescents aged 2–19 years exceed the national average and are higher than for other ethnoracial groups [2].

As adiposity increases in children and adolescents, so does the prevalence of cardiometabolic risk factors (CMRFs), including hypertension, dyslipidemia, and type 2 diabetes (T2D), all of which collectively make up metabolic syndrome [3]. The etiology of adiposity-based chronic disease and CMRFs includes ethnoracial disparities but these are rarely considered in current treatment approaches. As such, describing the prevalence, distribution, and factors associated with childhood obesity in more vulnerable groups may help to design effective interventions. Pragmatic real-world data from clinical settings that assess differences in the impact of adiposity on CMRFs in ethnoracially diverse adolescents are needed.

In Tennessee, rates of obesity among adolescents (20.8%) exceeded the national average [1], while 35% of adults were living with obesity in 2021 [4]. Compared with other US cities, Memphis, TN has the seventh highest proportion of African Americans (64.6%) but under-representation of Hispanics (7.7%) [3,5]. Lifedoc Health (LDH), a multidisciplinary healthcare organization, successfully implemented a data-driven model and clinical protocols to attenuate the burden of cardiometabolic-based chronic disease in children and adolescents in the Greater Memphis area [6]. However, the implementation of a more precise transcultural model requires the characterization of different CMRF phenotypes in high-risk minority groups. Thus, this study aims to elucidate disparities in the CMRFs of African American and Hispanic adolescents with varying levels of adiposity in a real-world setting.

## 2. Materials and Methods

### 2.1. Study Design and Sampling

A cross-sectional retrospective analysis was conducted of 3-year clinical data (2018–2020 ± 6 weeks) from the patient records of AA and H adolescents aged 12–17 years under LDH. Patients were included regardless of insurance coverage (private, Medicaid, or none) or whether they were assigned to the practice by the payer or referred from outside primary care for co-management [6]. Patients who were pregnant, underweight (body mass index-for-age percentile [BMI%] < 5.0%), or had any known genetic cause of obesity were excluded. All procedures were performed in accordance with the Declaration of Helsinki. Prior to data collection, informed consent was obtained. No independent ethics committee or institutional review board was sought, as this study does not meet the definition of human subject research as defined in Federal Regulation 45 CFR 46.102 [7].

### 2.2. Physical and Biochemical Parameters

Weight was measured in light clothing, without shoes, using a calibrated scale (Digital Platform Scale Pro Plus 2101KL, Health-o-meter^®^, McCook, IL, USA). Height was measured using a digital stationary stadiometer (Seca^®^ 264, Chino, CA, USA). Blood pressure (BP) was measured in the right arm, using appropriate cuff size, in a sitting position, with a aneroid sphygmomanometer (McKesson^®^, Irving, TX, USA) [8]. Blood glucose, glycated hemoglobin (HbA_1c_), high-sensitivity C-reactive protein (hs-CRP), fibrinogen, triglycerides (TG), total cholesterol (TC), high-density lipoprotein cholesterol (HDL-C), low-density lipoprotein cholesterol (LDL-C), aspartate aminotransferase (AST), alanine transaminase (ALT), and 25-OH-vitamin D were measured using random blood samples in a single certified laboratory (LabCorp, Burlington, NC, USA) [9].

### 2.3. Study Variables and Definitions

Participants self-reported as male or female and as African American and Hispanic. BMI% percentile was determined using age- and sex-based definitions of BMI, with individuals classified as having normal weight (<85th BMI%), overweight (≥85th and <95th BMI%), moderate obesity (≥95th and <99th BMI%), or severe obesity (≥99th BMI%) [10,11]. This modified version of the World Health Organization criteria (BMI% ≥95th) was used to define obesity due to limitations in using waist circumference (WC) to measure excess adiposity, including variability in cut-off points in age, race and ethnicity. Elevated blood pressure (BP) was defined for children aged 1–<13 years as ≥90th percentile or 120/80 mmHg (whichever was lower) and for children ≥13 years ≥120/<80 mmHg. Hypertension was defined for children aged <13 years as ≥95th percentile or ≥130/80 mmHg (whichever was lower) and for children ≥13 years as ≥130/80 mmHg or taking BP-lowering medication [12]. Dyslipidemia was defined as TC ≥ 200 mg/dL, LDL-C ≥ 130 mg/dL, TG ≥ 130 mg/dL, HDL-C < 40 mg/dL, or taking any lipid-lowering medications. Prediabetes was defined as HbA_1c_ ≥ 5.7% and <6.5%, and T2D as either HbA_1c_ ≥ 6.5% or a personal history of diabetes [13]. Fasting blood glucose was not used to define dysglycemia status due to practical difficulties in verifying the fasted state in a clinical setting. Metabolic syndrome (MetS) was defined as any ≥3 of the following criteria: overweight/obesity/severe obesity [14], elevated BP/hypertension, elevated TG, low HDL-C, and prediabetes/T2D [15], while elevated levels of liver enzymes, hs-CRP and fibrinogen, or vitamin D insufficiency were additional CMRFs [16,17]. High liver enzymes were defined as AST > 40 IU/L or ALT > 32 IU/L [9]. Vitamin D insufficiency was established if 25-OH-vitamin D was <30 ng/mL [18].

### 2.4. Statistical Analysis

Data were analyzed using R (version 3.6.2). Variables with normal distribution were presented as mean ± standard error, with differences between groups assessed by the student *t*-test. Frequencies were presented as percentages and 95% confidence intervals (CIs). Groups were compared using the χ^2^ test. Multivariate logistic regression models were applied to explore the relationships between exposures (age, sex, ethnoracial group, and adiposity) and outcomes (impaired glucose metabolism [IGM], elevated BP, high TC, low HDL-C, high LDL-C, high TG, high AST/ALT, inflammation, vitamin D insufficiency), adjusted by age and sex. Odds ratios and 95% CIs were estimated across different groups. The significance threshold was 0.05 for all analyses.

## 3. Results

### 3.1. Population Characteristics

Overall, the analysis cohort consisted of 2284 adolescents meeting inclusion criteria. Of these, 25.8% had private health insurance, 67.6% had a Medicaid healthcare plan, and 6.6% were uninsured. Overall, 687 (30.1%) were African American and 1597 (69.9%) were Hispanic, 50.3% were female, and mean age, weight, and BMI was 13.9 years, 70.9 kg, and 26.9 kg/m^2^, respectively. Sex distribution was similar between groups. Males had higher mean weight, height, BP, TG, blood glucose, AST, ALT, and vitamin D, while females had higher mean BMI, HDL-C, hs-CRP, and fibrinogen. African American adolescents had higher mean weight, height, BMI, BP, LDL-C, hs-CRP, and HbA_1c_, and lower vitamin D, while Hispanic adolescents had higher TG, AST, and ALT. Hispanic males had lower HDL-C and higher blood glucose than African American males (Table 1).

### 3.2. CMRF Prevalence by Ethnoracial Group, Sex, and Adiposity

Of the total cohort, 40.7% were a normal weight, 20.2% were overweight, 25.0% had obesity, and 14.2% had severe obesity (Table 2). Prevalence was high for the following CMRFs: vitamin D insufficiency, inflammation, dyslipidemia, MetS, high TG, low HDL-C, IGM, prediabetes, hypertension, high liver enzymes, elevated BP, high TC, high LDL-C, and T2D (92.4%, 55.6%, 46.2%, 34.1%, 29.2%, 24.6%, 18.0%, 16.3%, 12.3%, 12.0%, 8.5%, 7.1%, 4.9%, and 1.7%, respectively; Table 2). Males had a higher frequency of severe obesity, elevated BP, hypertension, dyslipidemia, high TG, low HDL-C, MetS, and high liver enzymes, while females had significantly higher rates of inflammation and vitamin D insufficiency (Table 2).

A higher proportion of Hispanic than African American males had overweight (20.7% vs. 12.6%) or obesity (27.6% vs. 17.7%), although severe obesity was more frequent in African American males (11.4% vs. 26.2%). IGM was also more frequent in African American males (primarily driven by prediabetes as T2D prevalence was similar between the groups). Hispanic males had higher prevalence of dyslipidemia, high TG, and high liver enzymes. Among females, overweight was more frequent in Hispanic than African American adolescents (25.9% vs. 13.8%), no difference in obesity was found, and severe obesity was more frequent in African American females (5.4% vs. 27.3%). Hispanic females also had a higher prevalence of dyslipidemia, high TG, and high liver enzymes, while African American females had a higher prevalence of prediabetes, T2D, elevated BP, hypertension, high LDL-C, and vitamin D insufficiency (Table 2).

In both groups, adiposity was the strongest determinant of cardiometabolic risk. CMRFs became more prevalent as severity of adiposity increased, except by high TG levels that were more prevalent in Hispanic adolescents regardless of level of adiposity (Table 3). As adiposity increased from overweight to severe obesity, Hispanic adolescents exhibited a significantly higher prevalence of dyslipidemia, lower HDL-C, and higher liver enzymes, with MetS prevalence also significantly higher in those with obesity and severe obesity. African American adolescents demonstrated a higher prevalence of elevated BP in the overweight category, prediabetes in the overweight to severe obesity categories, and T2D in the obesity category (Table 3).

### 3.3. Association between Demographics and BMI Categories with Cardiometabolic Abnormalities

Males were significantly more likely to have elevated BP, low HDL-C, high TG, and high liver enzymes, while inflammation and vitamin D insufficiency were more likely in females (Table 4). Older adolescents (aged 15–17 years) had 3.4-fold higher odds of hypertension and were more likely to have high TC, high LDL-C, and high liver enzymes than younger adolescents (12–14 years) (Table 4). Compared with Hispanic, African American ethnicity was associated with 3.0-fold greater odds of IGM, whereas Hispanic adolescents were more likely to have low HDL-C (1.7-fold higher odds), high liver enzymes (5.9-fold higher odds), and high TG (8.3-fold higher odds) than African American adolescents (Table 4).

All CMRFs tended to increase with adiposity, with a significantly increased risk of IGM, elevated BP, low HDL-C, high TG, and inflammation in overweight vs. normal weight individuals. The increased risk of high liver enzymes and vitamin D insufficiency was significant in adolescents with overweight vs. normal weight, while risk of high TC and high LDL-C became significant with severe obesity (Table 4).

## 4. Discussion

This study illustrates ethnoracial differences in the association of the progressive increase in adiposity with the CMRF profile of African American and Hispanic adolescents receiving care at LDH. The main finding of this study was that, independently of ethnoracial group, age, and sex, adiposity was the strongest determinant of cardiometabolic risk. Another relevant finding was that BMI% thresholds used in clinical practice were not sufficient at highlighting health implications in adolescents with adiposity in an appropriate amount of time. By the time individuals were submitted for evaluation (≥85th BMI%, classified as overweight) [14], the prevalence of IGM, elevated BP, low HDL-C, and high TG were already elevated and continued to progress with increasing adiposity. Established clinical guidelines recommend the evaluation and treatment of individuals only once they are overweight [14]. Thus, early identification of high-risk individuals in these minority groups is lacking and ultimately precludes the potential for earlier and more proactive interventions [19,20].

The rate of increase in obesity has been greater in children than in adults in many countries [21], The frequency of obesity in our study (39.2%) was higher than the 20.6% reported for US adolescents (12–19 years) in NHANES (2015–2016) [22]. In NHANES, no overall difference was found in the prevalence of obesity between Hispanic adolescents (25.8%) and African American [22] adolescents (22.0%), although obesity was more frequent in Hispanic than African American males (28% vs. 19%). We found that African American adolescents had a higher prevalence of severe obesity in both sexes, although overweight was more prevalent in Hispanic adolescents of both sexes. Our study is not a population-based sampling analysis like NHANES, but a representation of a multi-specialty clinical setting where most patients were assigned by insurance companies, which may explain these differences. Several factors may also account for the higher prevalence of obesity in our study. First, the sample was drawn from a predominantly minority population where the estimated prevalence of obesity exceeds the national average [1]. Moreover, subjects from low-income families that were uninsured or Medicaid-covered represented 74.2% of our sample and it is known that prevalence of obesity is higher among these groups [14,21,23]. Furthermore, LDH has implemented a diabetes risk stratification program, thus patients may have been referred from outside primary care for co-management [8]. Finally, the lifestyle impact of the COVID-19 pandemic could have exacerbated obesity in those evaluated in 2020 [14,24].

The overall obesity prevalence in our study (39.2%) was similar to that reported for adults in NHANES (2015–2016; 39.8%) [22]. A meta-analysis involving 200,777 subjects reported that approximately 80% of adolescents with obesity remain with obesity into adulthood [25]. Data from the Bogalusa Heart Study which followed 2392 children (aged 5–14 years) for 17.1 years showed that African American children with obesity were more likely to remain with obesity as adults (83%) than their non-Hispanic White counterparts (68%) [26]. The cumulative effect of multiple CMRFs associated with increased adiposity in childhood and adolescence may lead to adult cardiovascular disease (CVD). Earlier, more intensive interventions may be needed to ameliorate the impact of childhood obesity on adult cardiovascular outcomes [27].

Although only 1 ethnoracial disparity in CMRFs (high TG) was found in normal weight adolescents, more marked phenotypic differences were expressed as the degree of adiposity increased. Similar results have been reported in other studies using representative national samples of adolescents [28,29]. MetS is a cluster of risk factors for cardiometabolic diseases, including heart disease, stroke, and diabetes [30]. In NHANES (1999–2008), MetS in adolescents (n = 3385) was reported as 35.4% and 24.6% in boys and girls with Obesity, respectively, per CDC classification standards, compared to 0.8% and 1.7% in their normal weight counterparts [31]. Like NHANES, we found increases in the prevalence of MetS as severity of adiposity increased. We previously evaluated the prevalence of CMRFs in 759 adolescents and adults with overweight/obesity undergoing LDH’s Diabetes Risk Stratification Program. Findings showed that once obesity is present, the proportion of subjects with ≥3 CMRFs was similar regardless of age, suggesting that obesity similarly impacts the CMRF profile of adolescents and adults [32].

Dyslipidemia prevalence has been reported to be between 19–25% in US adolescents when defined as any abnormal lipid or apolipoprotein B [33]. We found a higher prevalence of dyslipidemia (46.2%), which was greater in Hispanic than African American adolescents. Compared with non-Hispanic White adolescents, both African American and Hispanic adolescents exhibit a higher degree of insulin resistance in response to increased adiposity [34,35], which has been linked to high TG and low HDL-C, referred to as metabolic dyslipidemia [36,37]. However, the NHANES (2001–2012) study showed that in African American adolescents the impact of obesity-induced insulin resistance is expressed distinctively and with fewer adverse lipid effects than in Mexican American adolescents [36,37]. In our study, African American subjects exhibited a lower baseline TG level and lower high TG prevalence compared with Hispanic subjects across all adiposity categories.

Metabolic-associated fatty liver disease (MAFLD) is associated with both metabolic dyslipidemia and progression to prediabetes and T2D [38]. A meta-analysis reported that MAFLD prevalence is higher in children with obesity than normal weight (34.2% vs. 7.6%), higher in males with obesity than females, and increases with severity of adiposity [39]. In our study, the prevalence of high liver enzymes was 12%, with a three-fold higher prevalence in males and a >two-fold higher prevalence in Hispanic subjects, regardless of sex. Although the NHANES (1999–2004) survey reported a lower prevalence of elevated ALT >40 U/L (3.6%) in adolescents aged 12–19 years and significantly higher prevalence in Mexican American adolescents (6.1%) than African American adolescents (2.3%) [40], another NHANES survey (1999–2010) in normal weight 12–18-year-old adolescents detected no ethnoracial differences in ALT levels after adjusting for sex, WC, and weight [41]. Similarly, our study found no ethnoracial differences in the prevalence of high liver enzymes in normal weight adolescents, although prevalence increased in Hispanic adolescents with increasing adiposity. The true prevalence of high liver enzymes could be even higher than reported here, since lower pediatric cut-offs for ALT (males, 25.8 U/L; females, 22.1 U/L) have been proposed [42]. 

Inflammation is another common pathophysiological risk factor for CVD [43]. In our study prevalence was higher among African American adolescents regardless of sex. We have previously reported that comparable levels of adiposity and insulin resistance result in a higher level of subclinical inflammation in African American adolescents than in non-Hispanic White adolescents [44].

The most prevalent CMRF was vitamin D insufficiency (92.4%), which was most frequent in African American females. Vitamin D deficiency has previously been reported as more prevalent in female adolescents than males [45], and NHANES data have consistently shown a higher prevalence of vitamin D deficiency in African American and Hispanic adolescents compared with non-Hispanic White adolescents [46]. Disparities between African American and Hispanic subjects may be attributed to several factors, including darker skin pigmentation, amount of sun exposure required to produce vitamin D [45], and higher adiposity.

There are some limitations to this study. This retrospective analysis is liable to residual confounders and causal inferences cannot be established. Since these data were drawn from a specific clinical setting, these results cannot be extrapolated to the general population. IGM was defined only using HbA_1c_, which can cause bias IGM estimates in African American subjects; however, it is currently used by the American Diabetes Association to define prediabetes [13]. Moreover, not all blood samples were collected in a fasted state, which introduces variations in the lipid profile. However, strengths of this study include its large sample size and conduct in a translational clinical practice designed to approach patients with cardiometabolic conditions under a standardized protocol for screening, management, and data collection. The study also used the same laboratory testing protocol and analytic methodology throughout the 3-year observation period regardless of patient payment capacity. Importantly, most adolescents were from low-income ethnoracial minority groups, who are usually under-represented in clinical trials.

## 5. Conclusions

The prevalence of overweight/obesity in the LDH system is very high among African American and Hispanic adolescents. Increased adiposity was the most impactful determinant of the CMRF profile in this population, even at lower levels of excess adiposity. The current BMI-based approach is not sensitive enough to allow the early identification of adolescents at higher risk, and differences between CMRF profiles in African American and Hispanic adolescents in response to increased adiposity are not addressed in current management guidelines, delaying the implementation of appropriate primary preventive measures. Research within routine clinical settings, particularly in minority groups, is required to characterize the nature of these ethnoracial discrepancies. This will better guide healthcare providers in more precise interventions to ameliorate any adverse health impact or underlying burden to the health system as they transition to adulthood.

## Figures and Tables

**Table 1 nutrients-16-03143-t001:** Cardiometabolic risk factors in adolescents by sex and ethnoracial group.

	Male (M)				Female (F)				Total	
	African American (AA)	Hispanic (H)	Total	*p* Value AA vs. H	African American (AA)	Hispanic (H)	Total	*p* Value AA vs. H	Total	*p* Value M vs. F
n (%)									2284 (100.0)	
Age, years			13.9 ± 0.05				13.9 ± 0.05		13.9 ± 0.04	
Weight ^a^, kg	82.2 ± 1.8	69.6 ± 0.7	73.3 ± 0.8	≤0.001	81.4 ± 1.5	62.2 ± 0.6	68.6 ± 0.7	≤0.001	70.9 ± 0.5	≤0.001
Height ^a^, cm	169.7 ± 0.6	163.3 ± 0.4	165.2 ± 0.3	≤0.001	161.8 ± 0.4	155.2 ± 0.2	157.4 ± 0.2	≤0.001	161.3 ± 0.2	≤0.001
BMI ^a^, kg/m^2^	28.2 ± 0.6	25.8 ± 0.2	26.5 ± 0.2	≤0.001	30.8 ± 0.5	25.6 ± 0.2	27.4 ± 0.2	≤0.001	26.9 ± 0.2	0.01
SBP, mmHg ^b^	112.5 ± 0.7	108.8 ± 0.4	109.8 ± 0.3	≤0.001	110.0 ± 0.6	104.3 ± 0.3	106.2 ± 0.3	≤0.001	108.0 ± 0.2	≤0.001
DBP, mmHg ^b^	70.0 ± 0.5	67.9 ± 0.3	68.5 ± 0.3	≤0.001	70.3 ± 0.5	65.2 ± 0.3	66.9 ± 0.3	≤0.001	67.7 ± 0.2	≤0.001
Total cholesterol, mg/dL	153.3 ± 2.1	154.7 ± 1.4	154.8 ± 1.1	0.58	156.4 ± 1.9	154.9 ± 1.21	155.4 ± 1.0	0.51	155.1 ± 0.8	0.70
LDL-C, mg/dL	88.2 ± 1.9	82.9 ± 1.1	84.8 ± 0.1	0.02	90.5 ± 1.8	83.0 ± 1.1	85.4 ± 0.9	≤0.001	85.1 ± 0.7	0.64
HDL-C, mg/dL	47.9 ± 0.9	44.9 ± 0.5	45.7 ± 0.4	≤0.001	50.5 ± 0.8	49.2 ± 0.5	49.6 ± 0.4	0.14	47.7 ± 0.3	≤0.001
TG, mg/dL	87.6 ± 4.0	139.8 ± 3.7	125.8 ± 3.0	≤0.001	78.3 ± 2.5	117.2 ± 2.7	104.5 ± 2.1	≤0.001	114.7 ± 1.8	≤0.001
hs-CRP, mg/dL	3.42 ± 0.4	2.54 ± 0.2	2.82 ± 0.2	0.04	4.96 ± 0.4	2.94 ± 0.2	3.79 ± 0.2	≤0.001	3.3 ± 0.1	≤0.001
Fibrinogen, mg/dL	316.3 ± 6.5	318.3 ± 3.8	318.5 ± 3.3	0.77	368.3 ± 6.1	355.2 ± 3.8	360.6 ± 3.3	0.07	338.9 ± 2.4	≤0.001
Blood glucose, mg/dL	95.7 ± 1.2	99.7 ± 0.7	98.6 ± 0.6	≤0.001	95.4 ± 1.1	95.1 ± 0.6	95.2 ± 0.6	0.82	96.8 ± 0.4	≤0.001
HbA_1c_, %	5.54 ± 0.03	5.38 ± 0.02	5.42 ± 0.01	≤0.001	5.56 ± 0.03	5.36 ± 0.02	5.43 ± 0.02	≤0.001	5.42 ± 0.01	0.55
AST, IU/L	23.4 ± 0.6	26.1 ± 0.5	25.4 ± 0.4	≤0.001	19.2 ± 0.4	20.9 ± 0.4	20.4 ± 0.3	≤0.001	22.8 ± 0.3	≤0.001
ALT, IU/L	20.3 ± 0.9	28.8 ± 1.0	26.6 ± 0.8	≤0.001	15.7 ± 0.8	18.8 ± 0.7	18.0 ± 0.5	≤0.001	22.2 ± 0.5	≤0.001
25-OH-Vitamin D, ng/mL	17.3 ± 0.6	19.4 ± 0.4	19.1 ± 0.3	≤0.001	15.0 ± 0.5	17.8 ± 0.4	16.9 ± 0.3	≤0.001	18.0 ± 0.2	≤0.001

Variables are expressed as mean ± standard error. Sex (male vs female for total sample) and ethno-racial group (African American vs Hispanic for each sex) groups were compared using *t*-test. The sample size for each variable differed between groups; therefore, the mean (standard deviation) was calculated based on observed numbers and not the total cohort. ^a^ These variables were not adjusted by adiposity. ^b^ SBP/DBP were adjusted by height (cm). Abbreviations: ALT, alanine transaminase; AST, aspartate aminotransferase; BMI, body mass index; DBP, diastolic blood pressure; HbA_1c_, glycated hemoglobin; HDL-C, high-density lipoprotein cholesterol; hs-CRP, high-sensitivity C-reactive protein; LDL-C, low-density lipoprotein cholesterol; SBP, systolic blood pressure; TG, triglycerides.

**Table 2 nutrients-16-03143-t002:** Prevalence of cardiometabolic risk factors, high liver enzymes, and vitamin D insufficiency in adolescents by sex and ethnoracial group.

		Male (M)				Female (F)				Total	
		African American (AA)	Hispanic (H)	Total	*p* Value AA vs. H	African American	Hispanic	Total	*p* Value AA vs. H	Total	*p* Value M vs. F
Overweight ^a^		12.6	20.7	18.4	≤0.001	13.8	25.9	21.8	≤0.001	20.2	0.05
		(9.3, 16.9)	(18.0, 23.7)	(16.3, 20.8)		(10.5, 17.8)	(22.9, 29.2)	(19.5, 24.3)		(18.6, 21.9)	
Obesity ^a^		17.7	27.6	24.8	≤0.001	25.7	24.6	25.1	0.75	25.0	0.92
		(13.7, 22.4)	(24.6, 30.9)	(22.4, 27.4)		(21.4, 30.5)	(21.7, 27.8)	(22.6, 27.7)		(23.2, 26.8)	
Severe obesity ^a^		26.2	11.4	15.8	≤0.001	27.3	5.4	12.5	≤0.001	14.2	0.03
		(21.5, 31.5)	(9.4, 13.9)	(13.8, 18.0)		(22.9, 32.2)	(4.0, 7.3)	(10.7, 14.6)		(12.8, 15.6)	
Abnormal BP	Elevated	13.2	9.3	10.4	0.08	9.4	5.2	6.6	0.02	8.5	≤0.001
		(9.6, 18.0)	(7.3, 11.6)	(8.7, 12.5)		(6.5, 13.2)	(3.8, 7.2)	(5.2, 8.3)		(7.3, 9.8)	
	HTN	15.8	14.6	14.9	0.69	17.2	6.2	9.7	≤0.001	12.3	≤0.001
		(11.8, 20.8)	(12.2, 17.3)	(12.8, 17.3)		(13.3, 21.9)	(4.6, 8.3)	(8.0, 11.7)		(10.9, 13.8)	
Abnormal HbA_1c_	Pre-DM	33.1	11.2	17.1	≤0.001	27.5	8.9	15.5	≤0.001	16.3	0.53
		(25.7, 41.4)	(8.3, 14.8)	(14.1, 20.5)		(21.4, 34.6)	(6.2, 12.5)	(12.6, 18.8)		(14.2, 18.6)	
	T2D	2.7	0.8	1.3	0.09	4.8	0.9	2.2	0.01	1.7	0.34
		(0.9, 7.2)	(0.2, 2.4)	(0.6, 2.7)		(2.3, 9.1)	(0.2, 2.7)	(1.2, 3.9)		(1.1, 2.7)	
	IGM	35.8	11.9	18.3	≤0.001	32.3	9.8	17.7	≤0.001	18.0	0.84
		(28.2, 44.1)	(9.0, 15.6)	(15.2, 21.8)		(25.8, 39.5)	(7.0, 13.5)	(14.6, 21.2)		(15.8, 20.4)	
Dyslipidemia		41.8	58.5	54.3	≤0.001	33.7	41.8	38.9	0.04	46.2	≤0.001
		(34.7, 49.2)	(54.1, 62.8)	(50.5, 57.9)		(28.0, 40.0)	(37.5, 46.2)	(35.5, 42.5)		(43.7, 48.8)	
High total cholesterol		8.5	7.2	8.1	0.7	6.8	6.2	6.2	0.86	7.1	0.19
		(5.1, 13.6)	(5.2, 9.9)	(6.2, 10.4)		(4.1, 10.9)	(4.3, 8.7)	(4.7, 8.2)		(5.9, 8.6)	
High LDL-C		5.3	4.9	5.2	0.98	8.0	3.3	4.7	0.01	4.9	0.76
		(2.7, 9.8)	(3.3, 7.3)	(3.7, 7.1)		(5.1, 12.3)	(2.0, 5.3)	(3.4, 6.5)		(3.9, 6.2)	
High TG		13.2	44.0	35.8	≤0.001	7.6	30.6	23.2	≤0.001	29.2	≤0.001
		(8.9, 19.1)	(39.7, 48.5)	(32.3, 39.5)		(4.8, 11.8)	(26.7, 34.8)	(20.3, 26.3)		(26.9, 31.6)	
Low HDL-C		26.5	32.9	31.4	0.12	18.9	18.2	18.3	0.89	24.6	≤0.001
		(20.4, 33.4)	(28.9, 37.2)	(28.0, 34.9)		(14.3, 24.4)	(15.0, 21.8)	(15.7, 21.2)		(22.4, 26.8)	
Inflammation		55.3	43.8	47.6	0.03	68.9	60.6	64.1	0.1	55.6	≤0.001
		(46.4, 63.9)	(38.2, 49.5)	(42.9, 52.3)		(61.1, 75.9)	(54.4, 66.6)	(59.4, 68.6)		(52.3, 58.9)	
Metabolic syndrome		35.8	42.6	41.1	0.18	30.1	24.3	27.0	0.18	34.1	≤0.001
		(28.2, 44.1)	(37.8, 47.7)	(37.0, 45.3)		(23.7, 37.3)	(20.0, 29.2)	(23.4, 30.9)		(31.3, 37.0)	
High liver enzymes		9.0	21.2	18.1	≤0.001	2.1	7.8	6.3	≤0.001	12.0	≤0.001
		(5.5, 14.3)	(17.8, 25.1)	(15.4, 21.1)		(0.8, 5.1)	(5.7, 10.6)	(4.7, 8.3)		(10.4, 13.7)	
Vitamin D insufficiency		93.5	90.6	90.8	0.32	96.9	92.5	93.9	0.04	92.4	0.04
		(88.4, 96.5)	(87.4, 93.0)	(88.2, 92.8)		(93.4, 98.6)	(89.6, 94.6)	(91.8, 95.5)		(90.8, 93.7)	

Frequencies are expressed as percentages and 95% confidence intervals. Groups were compared using the χ² test. ^a^ These variables were not adjusted for adiposity. Abbreviations: BP, blood pressure; HbA_1c_, glycated hemoglobin; HDL-C, high-density lipoprotein cholesterol; HTN, hypertension; IGM, impaired glucose metabolism; LDL-C, low-density lipoprotein cholesterol; pre-DM, prediabetes mellitus; T2D, type 2 diabetes; TG, triglycerides.

**Table 3 nutrients-16-03143-t003:** Prevalence of cardiometabolic risk factors, high liver enzymes, and vitamin D insufficiency by severity of adiposity (via BMI%) and ethnoracial group in the total sample.

**%**		**Normal Weight**						**Overweight**								
		**African American**		**Hispanic**		**Total, %** **(Normal Weight)**	***p* Value**	**African American**		**Hispanic**			**Total, %** **(** **Overweight)**		***p* Value**	
Abnormal BP	Elevated	4.9 (2.6, 8.8)		3.3 (2.1, 5.1)		3.7 (2.5, 5.2)	0.38	15.4 (8.5, 25.7)		5.9 (3.7, 9.1)			7.5 (5.3, 10.6)		0.01	
	Hypertension	6.2 (3.6, 10.4)		3.8 (2.5, 5.7)		4.4 (3.1, 6.0)	0.19	3.8 (1.0, 11.6)		6.2 (4.0, 9.5)			5.9 (3.9, 8.7)		0.59	
Abnormal HbA_1c_	Pre-DM	2.4 (0.1, 14.4)		2.3 (0.6, 7.0)		2.3 (0.7, 6.1)	1.00	15.2 (5.7, 32.7)		7.6 (4.5, 12.5)			8.6 (5.5, 13.1)		0.18	
	DM	NA		1.5 (0.3, 5.9)		1.1 (0.2, 4.5)	1.00	NA		0.5 (0.0, 3.2)			0.4 (0.0, 2.7)		1.00	
	IGM	2.4 (0.1, 14.4)		3.8 (1.4, 9.1)		3.4 (1.4, 7.6)	1.00	15.2 (5.7, 32.7)		8.1 (4.9, 13.1)			9.0 (5.8, 13.6)		0.2	
Metabolic syndrome		2.6 (0.1, 15.1)		1.5 (0.3, 6.0)		1.7 (0.5, 5.4)	0.55	12.1 (4.0, 29.1)		23.3 (17.8, 29.8)			21.8 (16.9, 27.7)		0.23	
Dyslipidemia		24.3 (17.0, 33.4)		30.1 (25.3, 35.4)		28.3 (24.3, 32.7)	0.29	20.0 (10.5, 34.1)		41.7 (35.7, 48.0)			38.1 (32.8, 43.7)		0.01	
High total cholesterol		6.1 (2.7, 12.6)		4.8 (2.8, 7.8)		5.0 (3.3, 7.5)	0.76	12.0 (5.0, 25.0)		5.8 (3.4, 9.6)			7.0 (4.5, 10.5)		0.12	
High LDL-C		5.2 (2.1, 11.5)		3.0 (1.5, 5.6)		3.5 (2.1, 5.7)	0.25	4.0 (0.7, 14.9)		4.3 (2.3, 7.7)			4.5 (2.6, 7.5)		1	
High TG		4.3 (1.6, 10.3)		20.9 (16.7, 25.7)		16.3 (13.1, 20.1)	≤0.001	2.0 (0.1, 12.0)		32.4 (26.8, 38.6)			27.6 (22.8, 33.0)		≤0.001	
Low HDL-C		10.4 (5.7, 17.9)		8.9 (6.2, 12.6)		9.3 (6.9, 12.5)	0.77	6.0 (1.6, 17.5)		20.5 (15.8, 26.0)			18.1 (14.1, 22.9)		**0.03**	
Inflammation		NA		11.1 (2.9, 30.3)		7.0 (1.8, 20.1)	0.54	17.4 (5.7, 39.5)		36.8 (29.3, 45.1)			34.1 (27.2, 41.7)		0.11	
High liver enzymes		1.9 (0.3, 7.5)		4.5 (2.6, 7.4)		4.0 (2.5, 6.4)	0.38	NA		8.9 (5.8, 13.3)			7.3 (4.7, 11.0)		**0.03**	
Vitamin D insufficiency		91.2 (82.3, 96.1)		85.1 (80.2, 89.0)		86.2 (82.1, 89.6)	0.22	95.2 (82.6, 99.2)		89.7 (84.8, 93.1)			90.3 (86.1, 93.4)		0.39	
**%**		**Obesity**					**Severe Obesity**							**Sample**		
		**African American**	**Hispanic**		**Total, %** **(** **Obesity)**	***p* Value**	**African American**		**Hispanic**		**Total, %** **(** **Severe Obesity)**	***p* Value**		**%**		***p* Value**
Abnormal BP	Elevated	8.7 (4.6, 15.3)	11.9 (8.9, 15.6)		11.8 (9.2, 14.9)	0.4	19.8 (14.1, 26.9)		16.3 (10.6, 24.0)		17.8 (13.7, 22.7)	0.54		8.5 (7.3, 9.8)		≤0.001
	Hypertension	20.5 (14.0, 28.7)	17.3 (13.7, 21.5)		18.0 (14.9, 21.6)	0.5	34.0 (26.8, 41.9)		33.3 (25.4, 42.2)		33.9 (28.6, 39.6)	1.00		12.3 (10.9, 13.8)		≤0.001
Abnormal HbA_1c_	Pre-DM	29.9 (21.6, 39.6)	11.4 (8.1, 15.6)		16.2 (12.9, 20.1)	≤0.001	40.4 (32.7, 48.5)		21.0 (13.9,30.2)		32.1 (26.7, 38.1)	≤0.001		16.3 (14.2, 18.6)		≤0.001
	DM	6.5 (2.9, 13.5)	0.3 (0.0, 2.1)		1.9 (0.9, 3.8)	≤0.001	3.8 (1.6, 8.6)		1.9 (0.3, 7.4)		3.0 (1.4, 6.0)	0.48		1.7 (1.1, 2.7)		0.17
	IGM	36.4 (27.5, 46.4)	11.7 (8.4, 15.9)		18.1 (14.6, 22.1)	≤0.001	44.2 (36.4, 52.4)		22.9 (15.5, 32.3)		35.1 (29.4, 41.1)	≤0.001		18.0 (15.8, 20.4)		≤0.001
Metabolic syndrome		29.8 (21.4, 39.7)	44.2 (38.6, 49.8)		41.1 (36.5, 45.9)	0.01	46.2 (38.3, 54.3)		64.2 (54.2, 73.1)		54.0 (47.9, 60.0)	0.01		34.1 (31.3, 37.0)		≤0.001
Dyslipidemia		38.4 (29.5, 48.1)	68.1 (62.7, 73.1)		60.4 (55.7, 65.0)	≤0.001	50.9 (43.0, 58.8)		77.8 (68.6, 85.0)		62.0 (56.0, 67.7)	≤0.001		46.2 (43.7, 48.8)		≤0.001
High total cholesterol		6.2 (2.8, 12.9)	7.1 (4.6, 10.5)		6.9 (4.8, 9.7)	0.94	8.1 (4.5, 13.7)		13.9 (8.2, 22.2)		11.1 (7.8, 15.5)	0.18		7.1 (5.9, 8.6)		0.02
High LDL-C		5.4 (2.2, 11.8)	3.1 (1.6, 5.8)		3.8 (2.3, 6.1)	0.26	10.0 (6.0, 16.0)		10.3 (5.5, 18.0)		9.7 (6.6, 14.0)	1.00		4.9 (3.9, 6.2)		≤0.001
High TG		8.9 (4.6, 16.2)	50.0 (44.6, 55.4)		39.1 (34.6, 43.8)	≤0.001	17.4 (12.1, 24.3)		61.1 (51.2, 70.2)		36.2 (30.6, 42.2)	≤0.001		29.2 (26.9, 31.6)		≤0.001
Low HDL-C		23.2 (16.0, 32.3)	39.1 (33.9, 44.7)		35.0 (30.7, 39.7)	≤0.001	34.8 (27.6, 42.7)		47.7 (38.1, 57.4)		40.0 (34.3, 46.0)	0.05		24.6 (22.4, 26.8)		≤0.001
Inflammation		55.0 (44.8, 64.9)	53.4 (47.5, 59.3)		54.1 (49.1, 59.1)	0.88	80.6 (73.4, 86.4)		77.7 (68.2, 85.0)		79.9 (74.4, 84.4)	0.67		55.6 (52.3, 58.9)		0.02
High liver enzymes		3.5 (1.1, 9.4)	20.9 (16.7, 25.8)		16.4 (13.2, 20.2)	≤0.001	9.9 (6.0, 15.9)		39.1 (30.1, 48.9)		22.4 (17.8, 27.8)	≤0.001		12.0 (10.4, 13.7)		≤0.001
Vitamin D insufficiency		94.5 (88.0, 97.8)	95.9 (93.0, 97.7)		95.2 (92.7, 97.0)	0.59	98.1 (94.2, 99.5)		98.2 (92.9, 99.7)		97.9 (95.2, 99.1)	1.00		92.4 (90.8, 93.7)		≤0.001

Frequencies are expressed as percentages and 95% confidence intervals, and groups were compared using the χ² test. Abbreviations: BMI, body mass index; BP, blood pressure; DM, diabetes mellitus; HDL-C, high-density lipoprotein cholesterol; HbA_1c_: glycated hemoglobin; IGM, impaired glucose metabolism; LDL-C, low-density lipoprotein cholesterol; NA, not available; TG, triglycerides.

**Table 4 nutrients-16-03143-t004:** Association between demographics and BMI categories with cardiometabolic abnormalities in the total sample (logistic regression analysis adjusted by age and sex).

		IGM	Elevated BP ^a^	High TC	Low HDL-C	High LDL-C	High TG	High AST/ALT	Inflammation	Vitamin D Insufficiency
Sex	Male	1	1	1	1	1	1	1	1	1
	Female	0.88	0.48	0.80	0.54	0.97	0.63	0.36	2.51	1.75
		(0.63, 1.23)	(0.39, 0.60)	(0.54, 1.20)	(0.42, 0.70)	(0.61, 1.56)	(0.50, 0.81)	(0.25, 0.52)	(1.85, 3.39)	(1.15, 2.65)
Age	12–14 years	1	1	1	1	1	1	1	1	1
	15–17 years	1.27	3.38	1.68	1.00	1.83	1.05	1.65	1.17	1.33
		(0.89, 1.81)	(2.69, 4.24)	(1.11, 2.54)	(0.75, 1.32)	(1.14, 2.95)	(0.79, 1.38)	(1.13, 2.40)	(0.83, 1.65)	(0.80, 2.19)
Ethnoracial group	Hispanic	1	1	1	1	1	1	1	1	1
	African American	3.01	1.13	0.87	0.58	1.16	0.12	0.17	0.80	1.30
		(2.09, 4.32)	(0.83, 1.53)	(0.54, 1.40)	(0.43, 0.79)	(0.68, 1.98)	(0.08, 0.18)	(0.19, 0.27)	(0.56, 1.15)	(0.75, 2.26)
BMI percentile	Normal weight	1	1	1	1	1	1	1	1	1
	Overweight	3.43	2.17	1.50	2.12	1.43	1.82	1.88	6.33	1.6
		(1.34, 8.79)	(1.56, 3.03)	(0.82, 2.76)	(1.39, 3.23)	(0.68, 2.99)	(1.27, 2.61)	(0.98, 3.61)	(1.85, 21.61)	(0.97, 2.65)
	Obesity	6.58	5.23	1.49	5.11	1.28	3.62	4.90	16.6	3.45
		(2.79, 15.52)	(3.92, 6.98)	(0.85, 2.59)	(3.54, 7.37)	(0.65, 2.53)	(2.61, 5.01)	(2.84, 8.46)	(4.98, 55.19)	(2.02, 5.90)
	Severe obesity	11.30	11.50	2.45	7.17	2.84	5.91	10.50	62.9	7.15
		(4.79, 26.82)	(8.35, 5.96)	(1.38, 4.38)	(4.77, 10.76)	(1.47, 5.49)	(3.94, 8.86)	(5.84, 18.81)	(18.39, 215.22)	(2.97, 17.23)

Data are odds ratios and 95% confidence intervals. Logistic regression analysis adjusted by sex and age was applied. Reference groups in the regression model were selected based on the lowest levels of perfect separation. ^a^ Elevated BP includes stage 1 and stage 2 hypertension. Abbreviations: ALT, alanine transaminase; AST, aspartate aminotransferase; BMI, body mass index; BP, blood pressure; HDL-C, high-density lipoprotein cholesterol; IGM, impaired glucose metabolism; LDL-C, low-density lipoprotein cholesterol; TC, total cholesterol; TG, triglycerides.

## Data Availability

Data are not publicly available due to legal restrictions on sharing/distribution of study data requiring Sponsor approval on a case-by-case basis. The data that support the findings of this study can be provided by the corresponding author (Velásquez) upon reasonable request.

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
