# Peer review of "Disparities in the Cardiometabolic Impact of Adiposity among African American and Hispanic Adolescents"

_nutrients, 2024, doi:10.3390/nu16183143_

Round 1

Reviewer 1 Report

Comments and Suggestions for Authors

1. Although this manuscript uses data derived from the Lifedoc Health (LDH) data, it should provide readers with enough information to ensure an autonomous understanding. Therefore, brief yet precise details should be provided on recruitment strategies and inclusion criteria; e.g., the current form of the manuscript does not explain to the reader why non-Hispanic White adolescents were excluded.

2. Lines 48-50: the definition of the metabolic syndrome is more nuanced than the association of any CMRF to excess weight; e.g., the metabolic syndrome does not include any form of dyslipidaemia, as mentioned below in lines 106-108.

3. Please explain the nature of the numerical variables in Tables 2 and 3 in the corresponding legends.

Author Response

  1. Although this manuscript uses data derived from the Lifedoc Health (LDH) data, it should provide readers with enough information to ensure an autonomous understanding. Therefore, brief yet precise details should be provided on recruitment strategies and inclusion criteria; e.g., the current form of the manuscript does not explain to the reader why non-Hispanic White adolescents were excluded.

The sample of white adolescents was too small to ensure reliable statistical analyses. To avoid confusion for the reader, we decided to exclude this group in the methodology (line 93) and results (lines 126 and 127). Seeing as the title already states an explicit focus on AA and H, there is no need to mention any other ethnoracial groups excluded from the sample. Thus, we will remove any mention of non-Hispanic White adolescents from the paper to avoid complicating the narrative. 

  1. Lines 48-50: the definition of the metabolic syndrome is more nuanced than the association of any CMRF to excess weight, e.g., the metabolic syndrome does not include any form of dyslipidaemia, as mentioned below in lines 106-108.

The purpose of lines 48-50 in the introductory paragraph is to state that increased adiposity is associated with other cardiometabolic risk factors collectively defining metabolic syndrome (MetS). We made a small modification to line 50 to improve clarity. However, the definition of MetS used in this study is detailed in lines 107-109 of the methodology. Elevated triglycerides (TG) and low high-density lipoprotein cholesterol (HDL-C) are included in the MetS definition, as proposed by Weiss

  1. Please explain the nature of the numerical variables in Tables 2 and 3 in the corresponding legends.

Done. Please, click twice on the tables to see all the table content and legends.

Reviewer 2 Report

Comments and Suggestions for Authors

This research aimed to characterize differences in CMRF among African American (AA) and Hispanic (H) adolescents with varying levels of adiposity. This topic is relevant to the field because it addresses an important issue, which is the cross-cultural disparity in CMRF phenotype. In a culturally diverse population like US, it is essential to the characterization of different CMRF phenotypes in high-risk minority groups. 

When comparing to previous literature, authors have comprehensive assessment on cardiovascular risk in population with diverse ethnicity, and has informed the necessity for early identification of high-risk individuals in minority groups. 

My minor comment is on the representativeness of samples, maybe comparing with NHANES data or other data in general population. Besides, it is unclear why vitamin D insufficiency data is deleted in Table 2. 

The conclusions are consistent with the findings. Authors have identified the most prevalent cardiometabolic disorder in each ethnic group, which agrees with the data presented in Tables. 

The references are appropriate.

Comments on the Quality of English Language

Nil.

Author Response

Reviewer : This research aimed to characterize differences in CMRF among African American (AA) and Hispanic (H) adolescents with varying levels of adiposity. This topic is relevant to the field because it addresses an important issue, which is the cross-cultural disparity in CMRF phenotype. In a culturally diverse population like US, it is essential to the characterization of different CMRF phenotypes in high-risk minority groups. 

When comparing to previous literature, authors have comprehensive assessment on cardiovascular risk in population with diverse ethnicity, and has informed the necessity for early identification of high-risk individuals in minority groups. 

My minor comment is on the representativeness of samples, maybe comparing with NHANES data or other data in general population. Besides, it is unclear why vitamin D insufficiency data is deleted in Table 2. 

The conclusions are consistent with the findings. Authors have identified the most prevalent cardiometabolic disorder in each ethnic group, which agrees with the data presented in Tables. 

The references are appropriate.

Response: Regarding comparisons to NHANES and sample representativeness, we acknowledge NHANES data in the paper as a reference for minority populations, particularly their disproportionate prevalence of obesity and associated conditions. However, our study is not a population-based sampling analysis like NHANES. It is a representation of a multi-specialty clinical setting where most patients were assigned to the practice by their insurance, with a higher-than-normal prevalence of obesity, overweight and associated conditions. We are not looking to be descriptive or draw conclusions about the general population of minorities, rather to characterize the impact of progressive adiposity on phenotypic differences in cardiometabolic risk profile among different ethnoracial groups.

As far as the Vitamin D Insufficiency Data missing from Table 2, the data is indeed in the table. Tables are simply embedded in the manuscript as a Microsoft Word Document Object and need to be double-clicked to be opened in a separate Word document for their full view. This is the case for all 4 of the tables in the manuscript, Table 2 just happens to be the only document too large to fit into the object preview that’s embedded in the main document.
